# Nanogold-Carried Graphene Oxide: Anti-Inflammation and Increased Differentiation Capacity of Mesenchymal Stem Cells

**DOI:** 10.3390/nano11082046

**Published:** 2021-08-11

**Authors:** Huey-Shan Hung, Mei-Lang Kung, Fang-Chung Chen, Yi-Chun Ke, Chiung-Chyi Shen, Yi-Chin Yang, Chang-Ming Tang, Chun-An Yeh, Hsien-Hsu Hsieh, Shan-hui Hsu

**Affiliations:** 1Graduate Institute of Biomedical Science, China Medical University, Taichung 40402, Taiwan; hunghs@mail.cmu.edu.tw (H.-S.H.); fish951753go@yahoo.com.tw (Y.-C.K.); hs0603@gmail.com (C.-A.Y.); 2Translational Medicine Research, China Medical University Hospital, Taichung 40402, Taiwan; 3Department of Medical Education and Research, Kaohsiung Veterans General Hospital, Kaohsiung 813414, Taiwan; kungmeilang@gmail.com; 4Department of Photonics, National Yang Ming Chiao Tung University, Hsinchu 30010, Taiwan; fcchen@mail.nctu.edu.tw; 5Neurological Institute Head of Department of Neurosurgery, Taichung Veterans General Hospital, Taichung 40705, Taiwan; shengeorge@yahoo.com (C.-C.S.); jean1007@gmail.com (Y.-C.Y.); 6Department of Physical Therapy, Hung Kuang University, Taichung 433304, Taiwan; 7Basic Medical Education Center, Central Taiwan University of Science and Technology, Taichung 40601, Taiwan; 8Collage of Oral Medicine, Chung Shan Medical University, Taichung 40201, Taiwan; ranger@csmu.edu.tw; 9Blood Bank, Taichung Veterans General Hospital, Taichung 40705, Taiwan; hhhsu@vghtc.gov.tw; 10Institute of Polymer Science and Engineering, National Taiwan University, Taipei 10617, Taiwan

**Keywords:** gold nanoparticles, graphene oxide, mesenchymal stem cells, anti-inflammation, differentiation

## Abstract

Graphene-based nanocomposites such as graphene oxide (GO) and nanoparticle-decorated graphene with demonstrated excellent physicochemical properties have worthwhile applications in biomedicine and bioengineering such as tissue engineering. In this study, we fabricated gold nanoparticle-decorated GO (GO-Au) nanocomposites and characterized their physicochemical properties using UV-Vis absorption spectra, FTIR spectra, contact angle analyses, and free radical scavenging potential. Moreover, we investigated the potent applications of GO-Au nanocomposites on directing mesenchymal stem cells (MSCs) for tissue regeneration. We compared the efficacy of as-prepared GO-derived nanocomposites including GO, GO-Au, and GO-Au (×2) on the biocompatibility of MSCs, immune cell identification, anti-inflammatory effects, differentiation capacity, as well as animal immune compatibility. Our results showed that Au-deposited GO nanocomposites, especially GO-Au (×2), significantly exhibited increased cell viability of MSCs, had good anti-oxidative ability, sponged the immune response toward monocyte-macrophage transition, as well as inhibited the activity of platelets. Moreover, we also validated the superior efficacy of Au-deposited GO nanocomposites on the enhancement of cell motility and various MSCs-derived cell types of differentiation including neuron cells, adipocytes, osteocytes, and endothelial cells. Additionally, the lower induction of fibrotic formation, reduced M1 macrophage polarization, and higher induction of M2 macrophage, as well as promotion of the endothelialization, were also found in the Au-deposited GO nanocomposites implanted animal model. These results suggest that the Au-deposited GO nanocomposites have excellent immune compatibility and anti-inflammatory effects in vivo and in vitro. Altogether, our findings indicate that Au-decorated GO nanocomposites, especially GO-Au (×2), can be a potent nanocarrier for tissue engineering and an effective clinical strategy for anti-inflammation.

## 1. Introduction

Organ and tissue failures and developmental defects can result from physical injury, genetic deficiency, aging, and diseases, and have become major social-economic and healthcare issues [1]. The challenges of traditional tissue engineering include a shortage of organ donors or lack of suitable organs and tissue, and limited time for matching and transport, which have all limited successful tissue and organ transplantations. More recently, tissue engineering and regeneration medicine have integrated multidisciplinary fields of sciences including biology, chemistry, engineering, medicine, pharmaceutical, material science, and nanoscience. Therefore, tissue engineering and regenerative medicine have achieved spectacular advancements to improve, cure, repair, and/or replace damaged and diseased tissues and organs [2,3,4,5]. To date, with the development of nanoscience, nanomaterials with extraordinary physicochemical properties have rapidly attracted tremendous interest and have been extensively applied in various fields including biosensors, theranostics, and biomedicine such as drug delivery, bioimages, and tissue engineering and regenerative medicine [6,7].

Stem cells (SCs) have been generally identified as (i) pluripotent stem cells including embryonic stem cells (ESCs) and inducible pluripotent stem cells (iPSCs) and (ii) multipotent adult stem cells such as mesenchymal stem cells (MSCs). As an extraordinary cell linage, stem cells are versatile and exhibit fundamental characteristics including self-renewability and specific differentiation to specialized cell types [4]. Among them, MSCs functionally possess long-term self-renewal capacity and can differentiate into multiple cell types including osteocytes, adipocytes, chondrocytes, endothelial cells, and neurons [8]. Due to the multilineage differentiated characteristics of stem cells, combined with nanomaterials for reprogrammed, stem cells have promoted the development of tissue engineering and regenerative medicine to a new milestone [5,9]. For example, previously studies have demonstrated that biopolymers such as collage and fibronectin incorporated gold nanoparticles (AuNPs) coated onto catheters can significantly elicit MSCs-endothelial differentiation and indicated this potential nanocomposite for regeneration of vascular tissue [10,11]. Furthermore, magnetic nanocomposite hydrogel (MNH) has been demonstrated to enhance the adhesion density of bone marrow MSCs for cartilage engineering [12].

Because of their large specific surface area, excellent physical, chemical, and mechanical properties, and superior biocompatibility, nanomaterials such as nanocomposites and metal NPs have been incorporated in polymeric scaffolds which are regarded as a potential and versatile platform for tissue engineering [13,14]. Furthermore, an ideal nanomaterial for tissue engineering must have an appropriate microenvironment to orchestrate with the extracellular matrix of the target tissue and/or multipotent stem cells conditions suitable for stimulating cell proliferation, cell adhesion, as well as guiding cell differentiation [15]. Graphene-based nanomaterials (GBNs) such as graphene, graphene oxide (GO), reduced graphene oxide (RGO), and modified graphene have unique physicochemical properties and good biocompatibility that contribute to extensive applications in biotechnology, biomedicine, bioengineering, disease diagnosis, and cancer therapy [7,16]. A single monolayer of graphene has a honeycomb lattice architecture with sp2 hybridized orbitals and tightly packed two-dimensional (2D) crystals [17]. Owing to its atomic thickness, large surface area, unique optical behavior, modifiable chemistry, and excellent thermal and electrical conductivity, graphene is considered to be a candidate for developing innovative theranostic tools. In addition, recent studies have indicated that graphene-derived nanomaterials have highly affinities for stem cell growth, maintenance, and differentiation [17,18,19]. GO is a graphene-derived oxidative form and a hydrophilic layered material. Moreover, structurally, GO plane contains functional groups including epoxide, hydroxyl, and carboxylic acid groups, as well as free surface π electrons which all contribute to GO sheets with characteristics of colloidal stability, polarity, and amphiphilicity [19,20]. In addition to graphene’s natural characteristics of mechanical, electrical, and thermal properties, the multi-chemical groups of GO sheets enable them to easily functionalize with either hydrophilic or hydrophobic molecules, drugs, and nanoparticles for biomedicine applications and tissue engineering [15,21,22,23].

An adequate substrate and/or scaffold for tissue engineering provides several critical properties such as mechanical support, chemical stimuli, and biological signals for induction of cell adhesion and proliferation [24,25]. The advantages of GO-Au include the abundant surface chemical groups on the GO and/or AuNP deposition as well as the enhanced surface topography, which are associated with cell adhesion and further cell proliferation. Indeed, surface features of nanocomposites including roughness, curvature, and wrinkled morphology all affect cell adhesion and proliferation. For example, 3D graphene foam with a wrinkled surface has been shown to induce neural cell growth rather than a smoother 2D graphene surface [26]. Moreover, nanotopographical network features on the surface of graphene nanogrid patterns have been demonstrated to successfully enhance osteogenic differentiation of MSCs [25].

Various biocompatible AuNPs and AuNP-derived nanocarriers have demonstrated efficacy and potential in biomolecular delivery [27], as well as anti-inflammatory effects in vascular tissue engineering [10,11,28]. The potent application of GO decorated with biopolymer for tissue engineering has been demonstrated with accumulating documents [29]. However, the efficacy of GO functionalized with metal nanoparticles such as silver and gold on stem cell-derived tissue engineering is still unclear and needs to be validated. In this study, we evaluate the effects of the as-fabricated AuNP-decorated GO nanocomposites for the biocompatibility and cell adherent efficiency of MSCs.

Here, we fabricate environmentally friendly and functional AuNP-deposited graphene oxide nanocomposites (Au-GO), characterize the feasibility of Au-GO nanocomposites on the anti-inflammatory effect in MSC-based tissue engineering, and evaluate the potent effect of these nanocarriers in tissue regeneration.

## 2. Materials and Methods

### 2.1. Synthesis of GO-Au Nanocomposites

Graphene oxide (GO) and HAuCl_4_ and sodium citrate were purchased from Sigma-Aldrich (Burlington, MA, USA). The synthesized procedure of GO-Au followed the Hummers’ method, as in previous reports [30,31]. The GO-Au solution was prepared by mixing GO (0.275 mg/mL, 3 mL) and HAuCl_4_ (0.047 mg/mL, 10 mL) components, and then aged for 30 min. After heating the solution to 80 °C, sodium citrate (0.085 mol/dm^3^) was added to reduce gold ions for 4 h of reaction. Then, the as-prepared suspensions were centrifuged (6000 rpms) and washed with ddH_2_O water twice to remove the unreacted gold ions. Ultimately, the lyophilization was processed to acquire dry GO-Au nanocomposites. 

### 2.2. Characterization Methods

The UV-Vis absorption spectra of as-prepared nanocomposites were analyzed using a UV-Vis spectrophotometer (Helios Zeta, Thermo Fisher, Pittsfield, MA, USA) with a 1 cm path length quartz cell. The spectrophotometer measures the range of waves from 200 to 1000 nm, where the peak at 520 nm is the absorption wavelength of AuNPs. Data was further analyzed and quantified using Origin Pro 8 (Originlab Corporation, Northampton, Massachusetts, USA) analysis software.

The transmission Fourier transform infrared (FTIR) spectra were acquired using a FTIR spectrometer (Shimadzu Pretige-21, Kyoto, Japan). The Fourier transform infrared (FTIR) spectra were acquired (8 cm^−1^ resolution, 256 scans, with the sample compartment vacuum pressure set at 0.12 hPa) using a Bruker 66 v/s FTIR spectrometer (Shimadzu Pretige-21, Kyoto, Japan). Double sided polished silicon (100) wafer substrates were cut into 20 × 20 mm^2^ pieces with a diamond-tipped stylus. Spectra from a freshly plasma-cleaned silicon wafer sample were collected before each measurement to obtain the background spectra. 

The surface morphology of the GO films was obtained using a Digital Instruments Dimension 3100 atomic force microscope (Symcon Inc., Charlotte, NC, USA).

The as-prepared nanocomposites were loaded on silicon substrates and a 0.7 μL of distilled water was dropped onto the surface of nanocomposites. Water contact angles of these nanomaterial were further determined using a PGX model instrument (DKSH Australia Pty Ltd, Braeside, VIC, Australia) at room temperature. 

### 2.3. Free Radical Scavenging Assay

The free radical scavenging potential of the as-prepared nanocomposites of GO, GO-Au, and GO-Au (×2) was achieved using a DPPH assay (2,2-diphenyl-1-picrylhydrazyl) (DPPH, Sigma-Aldrich, Burlington, MA, USA). The nanocomposites were dissolved in deionized water. To determine the antioxidant potential of these nanocomposites, the nanocomposites (1 mL) were mixed with DPPH at a ratio of 1:3, and then incubated in the dark at RT for 90 min. Then, the absorbance of the reaction mixture was measured at 539 nm using an ultraviolet-visible spectrophotometer (Helios Zeta, Thermo, Waltham, MA, USA). Free radical scavenging ability effect is determined by the following equation: % Inhibition = (1 − (A sample/A contol)) × 100, where A sample and A control are the absorbance value of the control (deionized water) and nanocomposites, respectively.

### 2.4. Cell Culture 

Wharton’s jelly-derived mesenchymal stem cells (MSCs) were harvested and isolated from human umbilical cord and kindly gifted from Prof. Woei-Cherng Shyu [32]. Briefly, cells were cultured in Dulbecco’s modified Eagle’s medium (DMEM) contained with 10% FBS, 1% sodium pyruvate, and 1% antibiotics (100 U/mL penicillin/streptomycin). 

### 2.5. Characterization of Mesenchymal Stem Cells 

By characterization of MSCs, the specific surface markers of MSCs were characterized by flow cytometry [32]. Briefly, cells were detached, washed, and then incubated with the indicated antibody conjugated with fluorescein isothiocyanate (FITC), phycoerythrin (PE), and/or PerCP-Cy5.5, against the indicated markers: CD14-FITC, CD29- FITC, CD34-FITC, CD44-PE, CD45-FITC, CD73-PE, CD90-FITC, and CD105-PerCP-Cy5.5 (BD Pharmingen, San Diego, CA, USA). PE-conjugated IgG1 and FITC-conjugated IgG1 were used as isotype controls (BD Pharmingen). Next, the antibody conjugated cells were analyzed using FACS analysis (Becton Dickinson LSR II, Canton, MA, USA). Cells of the 8th passage were used in this study. To detect the effects of as-prepared nanocomposites on cell cycle, MSC cells were seeded onto cover slides deposited with GO, GO-Au, and GO-Au (×2) for 24 h and harvested for flow cytometry analysis. The cell cycle distribution was determined using a propidium iodide (PI) staining (Sigma-Aldrich, Burlington, MA, USA). In brief, after cells were fixed with 80% ethanol, permeabilized with 0.01% Triton X-100, stained with PI solution (0.05 mg/mL of PI and 1 mg/mL of RNase A in PBS), and finally subjected to the FACS Caliber cytometer. The cells with sub-diploid DNA content were considered to be apoptotic cells. In both assays, 10,000 cell events were collected and analyzed with a flow cytometer (LSR II, BD Biosciences, Canton, MA, USA) and Flowjo 7.6 software (Tree Star, Ashland, OR, USA) analysis.

### 2.6. Cell Viability Assay

The cytotoxic effects of the as-prepared nanocomposites were detected using PI staining and calcein-AM assay, according to the manufacturer’s instructions. Briefly, cells (1 × 10^4^ cells/well) were seeded in 12-well culture plate which contained cover slides deposited with GO, GO-Au, or GO-Au (×2) for 24 h. Next, cells were incubated with PI (10 μg/mL) and/or calcein-AM solution (2 μM) for 30 min at 37 °C, and the cell viability was analyzed through detection of the fluorescence intensity using a fluorescent microscopy (ZEISS AXIO Z1, ZEISS, White Plains, NY, USA). Image J software (Media Cybernetics Inc Rockville, MD, USA) was used to quantify the intensity of fluorescence positive cells.

### 2.7. Monocyte and Platelet Activation Assay

The effects of as-fabricated nanocomposites for monocyte activation can be achieved by using the Percoll (Sigma-Aldrich, Burlington, MA, USA) isolation method and immunofluorescence assay, which has been described in a previous study [33]. Briefly, the isolated monocytes (1 × 10^5^ cells) were seeded onto the as-prepared nanocomposites and incubated at 37 °C for 96 h. Then, cells were subjected to immunofluorescent staining with a macrophage phenotypic marker of anti-CD68 antibody (GeneTex Inc., Irvine, CA, USA). Moreover, the numbers ratio of monocytes converted into macrophages (the conversion ratio) was indicated as an inflammatory index. For detection of platelet activation, platelet-rich plasma (2 × 10^6^ platelets/mL) was incubated with the as-prepared nanocomposites for 1 h, and then the plasma was removed. Then, the adherent platelets were counted and recorded using a cell counter (Metasizer, Coulter, IN, USA). Next, the platelet morphology was observed using scanning electron microscope (SEM) analysis according to the standard procedures. The platelet activation was identified and calculated using a standard formulation, as described in a previous study [34]. Briefly, the platelet activation was routinely defined into five stages according to their morphological change: 0 (round and inactivated), 0.25 (dendritic), 0.5 (spread-dendritic), 0.75 (spreading), and 1.0 (fully spread and totally activated). The average degree of platelet activation (0.0–1.0) was calculated with 50 adherent platelets based on SEM observations.

### 2.8. Determination of Intracellular Reactive Oxygen Species Generation

The cells (2 × 10^5^ cell/well in a 6-well culture plate) were seeded onto cover slides deposited with GO, GO-Au, and GO-Au (×2) for 48 h. To analyze the effects of as-prepared nanocomposites on intracellular ROS generation, the cells were harvested, and then incubated with an oxidation-sensitive fluorescent dye of 2′,7′-dichlorofluorescein diacetate (DCFH-DA) (10 nM, Sigma-Aldrich, Burlington, MA, USA) for 30 min at 37 °C. Next, the ROS production was determined using a flow cytometry (LSR II, BD Biosciences, Canton, MA, USA). The fluorescein-positive cells were evaluated using the FCS software (Becton Dickinson, Canton, MA, USA).

### 2.9. Enzyme-Linked Immunosorbent Assay (ELISA Assay)

The cells (1 × 10^4^ cell/well in a 12-well culture plate) were seeded onto cover slides deposited with GO, GO-Au, and GO-Au (×2) for 48 h. The supernatants were harvested, and the cytokines SDF-1 levels were measured using a human SDF-1 ELISA kit DuoSet^®^ ELISA Development System (R&D, Minneapolis, MN, USA), according to the manufacturer’s instructions. All analyses were carried out in duplicate. Each experiment was repeated four times. 

### 2.10. Metalloproteinases Zymography Assay

The effects of as-prepared nanocomposites on gelatin zymography were achieved according to a previous study [33]. Briefly, the cells (2 × 10^5^ cells) were seeded onto cover slides deposited with GO, GO-Au, and GO-Au (×2) for 48 h. Culture medium was collected to execute the gelatin zymography assay. The clear areas digested by MMP proteases on the gel were digitized using scanning densitometer and the MMPs’ digested bands were further quantified using Image Pro Plus 5.0 software (Media Cybernetics, Rockville, MD, USA). 

### 2.11. Immunofluorescence Staining Assay 

The cells (1 × 10^4^ cell/well in a 12-well culture plate) were seeded onto cover slides deposited with GO, GO-Au, and GO-Au (×2) for 24 h. Then, cells were fixed with 4% paraformaldehyde, permeabilized with 0.5% Triton-X 100 (Sigma-Aldrich, Burlington, MA, USA), and blocked with 5% BSA, sequentially. For observation of the cell adhesion, migration, and cytoskeletal morphology, cells were, next, incubated with anti-vinculin antibody and/or anti-CXCR4 antibodies (Santa Cruz, TX, USA) at 4 °C for overnight, and then stained with the F-actin rhodamine phalloidin (Sigma-Aldrich, Burlington, MA, USA) for 30 min. In addition, to detect the effects of nanocomposites on MSC differentiation, cells were incubated with antibodies including endothelial biomarkers (anti-CD31 and anti-vWF antibodies (Santa Cruz)), as well as neuronal biomarkers (anti-Nestin, anti-GFAP and anti-β-tubulin antibodies (Santa Cruz)) for overnight at 4 °C. Cell nuclear was stained with DAPI (Invitrogen, White Plains, NY, USA) for 10 min. After the cover slides were washed, mounted, and then applied to the fluorescence microscope (ZEISS AXIO IMAGER A1, White Plains, NY, USA) to capture the images processed in a dark room. The fluorescence positive cells were identified and quantified for the signal intensity using: Image Pro Plus 5.0 software (Media Cybernetics, Burlington, MA, USA). 

### 2.12. Alizarin Red S (ARS) Staining

MSC osteogenic differentiation was achieved using an ARS staining assay. Cells (1 × 10^4^ cells) were plated onto cover slides deposited with GO, GO-Au, and GO-Au (×2) for 7 days. The cells were fixed with 4% paraformaldehyde for 20 min. After deionized water washing, cells were stained with ARS (Sigma-Aldrich, 0.5% in deionized water, pH 6.3–6.4) for 45 min at room temperature. The calcium deposit on the MSC-differentiated osteocytes was observed with orange and red spots by using an inverted microscope. For staining quantification, calcium deposits stained with ARS were eluted with leaching solution (20% methanol and 10% acetic acid in distilled water) for 15 min. The absorbance of the destained supernatant was measured at 450 nm using the microplate reader.

### 2.13. Oil Red O (ORO) Staining Assay 

The MSCs (1 × 10^4^ cells) were plated onto cover slides deposited with GO, GO-Au, and GO-Au (×2) for 7 days. The cells were fixed with 4% paraformaldehyde for 20 min, rinsed with 60% isopropanol, and stained with ORO (Sigma-Aldrich, 0.35% in isopropanol) and hematoxylin for 10 min. Next, cells were washed twice with deionized water and dried at room temperature. Sample images were taken by fluorescence microscopy. To evaluate adipocyte differentiation, the number of ORO-positive adipocytes was counted and recorded from a given area on the culture slides.

### 2.14. In Vivo Biocompatibility Assay

Female Sprague Dawley (SD) rats (2–3 months old) were purchased from the National Laboratory Animal Center, Taiwan. All animal care and experimental procedures were performed according to the Institutional Animal Care and Use Committee (IACUC) of China Medical University and ensured that all animals received humane care and that study protocols complied with the institution’s guidelines. The materials were subcutaneously inoculated into a dorsally incised area of 10 mm^2^ of Sprague Dawley (SD) rats (300~350 g) under anesthesia and allowed to implant for 4 weeks. The animals (the numbers of every subgroup were n = 6) were sacrificed, and then the tissues surrounding the implanted nanocomposites were subjected to histological analysis and immunohistochemistry assay. By Hematoxylin and Eosin (H&E) staining, the implanted nanocomposites surface was calculated as the thickness of the fibrous capsule over 6 sites using Image J software to quantify the average encapsulated fibrotic tissues. To examine nanocomposites-mediated collagen deposition, the implanted nanocomposites sections were further executed to Masson’s trichrome staining assay kit (Sigma-Aldrich) for identifying collagen (blue color) from the surrounding tissue. The areas of fibrosis tissues were calculated using Image J 4.5 version software (Media Cybernetics). The pixels signaling strength of each fibrosis area from three randomly selected high-power fields (HPFs) were collected for further quantification. To evaluate the activation levels of M1 and M2 macrophage, the implanted nanocomposites sections were subjected to incubation with monoclonal anti-CD86 and anti-CD163 antibodies (1:200 dilution) (Dallas, TX, USA) respectively, and the secondary antibody of AF488 donkey anti-mouse IgG (Invitrogen, Carlsbad, CA, USA) and anti-mouse Immunoglobulin G (rhodamine) (1:500 dilution) (Jackson Immuno Research, Carlsbad, CA, USA) were used to detect protein signaling. The fluorescence intensity of CD86 and CD163 was further determined using an Olympus IX71 fluorescence microscope (Tokyo, Japan). All data are presented as the mean ± standard error of the mean (SEM). GraphPad Prism 5.0 (Graph Pad Software, La Jolla, CA, USA) was used for statistical analysis. Statistical analysis was performed with t-test statistical analysis. 

### 2.15. Statistical Analysis

All experiments independently repeated at least three times and data were expressed as mean ± standard deviation. Student’s t-test and SPSS Statistics v17.0 method were used to examine the difference between groups. The *p*-values less than 0.05 (*p* < 0.05) were considered statistically significant.

## 3. Results and Discussion

### 3.1. Characterization of Physicochemical Properties of Au-Decorated GO Nanocomposites

To validate the decoration of AuNPs onto GO, an UV-Vis spectroscopy was used to measure the absorption wavelength of the AuNPs. Our results showed that a typical peak at 520 nm for pure AuNP decoration was observed in the Au, GO-Au, and GO-Au (×2) groups (Figure 1A). By using FTIR analyses, as shown in Figure 1B, the spectra of GO revealed broad absorption bands near 3328.8 cm^−1^ and 1628.3 cm^−1^, which were attributed to the stretching mode of O–H bond [35] and C-O vibration, respectively [36]. The naked AuNPs displayed signals at 3462.7 cm^−1^ and 1635.6 cm^−1^ [37]. The strong peak at around 3400 cm^−1^ is due to the stretching mode of –OH [38]. When AuNPs were incorporated onto GO, the shifted adsorption peaks at 1642.8 cm^−1^ (GO-Au) and 1632.0 cm^−1^ ((GO-Au (×2)) were observed. The new peak located at 1425 cm-1 of GO-Au and GO-Au (×2) was ascribed to the carboxyl O=C–O or –OH bond from the –COOH sensing group in the functionalized GO-Au [38,39]. 

For pure GO, the concentration was approximately 0.01 mg mL^−1^. The AFM image is shown below. We observed that the GO covered more than 80% of the surfaces. The average size of the GO flakes was 2 μm, and the calculated surface concentration was 2 × 10^3^ cm^−2^. Without Au nanoparticles, the average thickness of the GO planes was 1.5 nm, which is the value for one or two layers of GOs (Figure 1C). The shape of the AuNP was very close to spherical and the average size of the AuNPs was 7 nm. The distribution of AuNPs was somehow not even. We found that more AuNPs formed on the edges and at the wrinkles, presumably because of the higher number of defect sites on these areas [30]. Next, we validated the surface hydrophilicity of these as-fabricated Au-decorated GO nanomaterials through water contact angle measurement. Our data indicated that the water contact angles of GO, GO-Au, and GO-Au (×2) were 88.5 ± 1.03°, 68.2 ± 2.04°, and 59.86 ± 2.36°, respectively (Figure 1D). Owing to surface hydrophilicity, wettability is one of the potential factors to regulate cell behavior via protein adsorption [40]. The contact angles of these nanomaterials were all less than 90 degrees, which indicated the liquid wets the surface and suggested their increasing hydrophilicity property is significantly associated with AuNP decoration. Moreover, the higher hydrophilicity of Au-decorated GO materials also implied that they may have affinity for cell adhesion. In addition, we also tested the antioxidant capacity and reactive oxygen species (ROS) scavenging efficiency of these three as-prepared nanocomposites, by using a free radical scavenging assay; we found that Au-deposited GO significantly enhanced the ROS scavenging and achieved 65% and 95% in GO-Au and GO-Au (×2), respectively, as compared with 40% in GO (Figure 1E). This result suggests that Au-deposited GO nanocomposites, especially GO-Au (×2), have excellent antioxidant property. Before these nanomaterials were subjected to experiments, we detected whether these as-prepared nanocomposites could be effectively stabilized on the glass substrates. By using GO nanocomposites as an example, GO was dissolved in either DI water or DI water/methanol mixture solution (1:1 in *v/v*), and then spin-coated onto glass slides. Following the GO-coated slide incubation with culture medium and/or saline buffer, we found that the group of GO dissolved in DI water/methanol mixture generated well-coated and stabilized GO film substrates (Figure 1F).

### 3.2. Cytocompatibility of Go-Au in MSCs 

First, we characterized the MSC phenotypes by detecting several surface markers [41] using a FACS analysis, including: (a) negative surface makers such as CD14, CD34, and CD45, which were expressed in hematopoietic cells, endothelial cells, and immune cells respectively, and (b) the minimum criteria surface antigen of MSCs CD29, CD44, CD73, CD90, and CD105 (Appendix A). The data from the FACS analysis further showed that less than 2% of negative markers (Appendix A) and more than 98% of the positive markers (Appendix A) were quantitated and validated, respectively, in MSCs. This result identified the “stemness” markers of the genuine MSCs that were, then, executed in subsequent experiments. Next, we verify the effects of these as-fabricated nanocomposites on the MSCs’ biocompatibility using a calcein-AM/PI co-staining assay (Figure 2A). By calcein-AM staining assay, cells displayed good viability on these nanocomposites with fluorescent intensity in a two-fold, three-fold, and four-fold in the GO, GO-Au, and GO-Au (×2), respectively, as compared with the control group (Figure 2B). Conversely, by PI staining assay, all three nanocomposites revealed less rare cell death as compared with the control group (Figure 2C). Furthermore, the steady step for cell viability is determined with their effective adherence ability toward the substrates. Then, we verified the effects of materials for MSC adherence efficiency. As shown in Figure 2D, cells were grown on these nanocomposites for 24 h, and then we characterized the expression and distribution of the representative cytoskeletal proteins including F-actin and vinculin by using an immunofluorescence assay. As compared with the control group, all three nanocomposites revealed robust cell attachment and morphological unwrapping. By further quantification of fluorescence intensity of vinculin (Figure 2E) and F-actin phalloidin (Figure 2F), showed that these nanocomposites had excellent adhesion capacity, greater than three-fold for the MSCs as compared with the control group. These results suggest that these Au-deposited GO nanocomposites have good biocompatibility and cell adhesion for MSCs. Therefore, both GO-Au and GO-Au (×2) mediated a significant increase in cell growth and good cell spreading for MSCs, which may be attributed to AuNP deposition changing the surface structure and/or texture of the nanomaterial substrate. 

### 3.3. Anti-Inflammatory Effect of GO-Au 

One of the challenges for nanocomposites subjected to biomedical applications is how to prevent them from eliciting severe immune responses and/or inflammation in the biomodels. Since the as-fabricated Au-deposited GO nanocomposites were found to significantly enhance ROS scavenging (Figure 1D), next, we detected the effects of these as-fabricated nanomaterials on immune activation. Human monocytes were isolated, and then incubated with the as-prepared nanomaterials for an indicated time course. The immune activation was further evaluated by analyzing the ratio of monocyte-to-macrophage differentiation. As shown in Figure 3A, after monocytes were cultured with nanomaterials for 96 h, the cells were examine for expression of CD68 marker in macrophages using an immunofluorescent assay. By counting the number of CD68-expressed cells, we found that GO-Au (×2) mediated a significantly lower activation of macrophage (Figure 3B) and reduced monocyte-to-macrophage conversion (Figure 3C) as compared with other groups. Moreover, we validated the effects of these nanomaterials on platelet activation using an SEM analysis. Our results indicate that nanomaterials significantly elicit dendritic and/or round-like morphological formations as compared with the well-spread morphology of the control group (Figure 3D). As calculated, regarding the platelet adherence efficiency, all nanomaterials revealed poor adherence capacity (Figure 3E) and a lower degree of platlet activation (Figure 3F) as compared with the control group. These results also suggest that the Au-deposited GO, especially GO-Au (×2), significantly suppresses platelet activation. Consistently, by using DCFH-DA staining and flow cytometry analysis, we also observed that the as-prepared nanomaterials showed a noticeable inhibition of ROS production in MSCs. Among that, GO-Au (×2) maintained a dramatic anti-oxidative efficacy as compared with the other groups (Appendix A). Altogether, these results suggest that Au-deposited GO nanocomposites, especially GO-Au (×2), have excellent anti-oxidative efficiency and lower induction of immune response. 

Further biocompatibility validation was evaluated by detecting cell viability using annexin V/PI staining assay (Figure 4A). Our results indicated that higher cell viability and lower cell death were observed in GO-Au (×2) group (Figure 4B,C). Reduced apoptosis was also found in both Au-decorated GO nanocomposites (Figure 4D). Next, we evaluated cell cycle distribution using a FACS analysis. Our results indicated that all three nanocomposites elicited fewer cell apoptosis than the control group, as indicated in the sub-G1 analysis. Moreover, the GO-Au (×2) group showed a significant increase in both the S phase and G2/M phase, and the other two nanocomposites revealed no different changes in the cell cycle distribution as compared with the control group (Figure 4E). These results suggest that Au-decorated GO nanocomposites are friendly and biocompatible for MSC growth and incubation. 

The biocompatible and non-toxicity of the versatile AuNPs have been well demonstrated in our previous study and numerous studies [42,43]. In this study, we fabricated Au-deposited GO nanocomposites and verified their biocompatibility, higher antioxidant efficacy, and lower ROS induction for MSCs. Moreover, we also demonstrated that the Au-deposited GO nanocomposites elicited lower activation of immune response including inhibition of monocyte transferring to macrophage differentiation and a decrease in platelets activation. Furthermore, GO exhibited its immense potential as an implantable material in guiding cell differentiation and enhancing cell growth of bone marrow-derived mesenchymal stem cells for cardiac repair and induction of osteogenic lineage [44,45]. Accordingly, these results suggest that Au-deposited GO nanocomposites may be a potent nanomaterial for tissue engineering. 

### 3.4. Enhanced Differentiation Ability of MSCs by GO-Au

Due to the potent efficacy of MSCs in revascularization of injured, ischaemic cardiomyopathy, and tissue regeneration [46], next, we evaluated the activation of the SDF-1/CXCR4 axis during Au-deposited GO-mediated MSC migration, tissue regeneration, and repair. By co-staining with CXCR4 and F-actin phalloidin using immunofluorescent assay, both GO-Au nanocomposites significantly elicited well-spread MSCs and high CXCR4 expression (Figure 5A). Further quantitated CXCR4 fluorescent intensity also suggested that CXCR4 was dramatically upregulated in both GO-Au and GO-Au (×2) nanocomposites with a 6.7- and 6.9-fold as compared with the GO (1.8-fold) and the control group (Figure 5B). By detecting the ligand for CXCR4, a small molecule chemokine-SDF-1α (stromal-derived factor-1α) was analyzed using an ELISA assay. Our results revealed that both GO-Au and GO-Au (×2) significantly elicited an increase in SDF-1α of 1.7- and 2.1-fold, respectively (Figure 5C). For cell migratory analysis, we examined the effect of these nanocomposites on MMP activity using a zymography activity assay. As shown in Figure 5D, these GO-Au nanocomposites, especially GO-Au (×2), significantly elevated both the MMP2 and MMP9 activity. This data indicated that these GO-Au nanocomposites significantly enhanced the WJ-MSC’s migration and motility. 

CXCR4 receptor can express in numerous sources of cell types including (i) the pluripotent/progenitor cells (haematopoietic stem cells, embryonic stem cells, and endothelial progenitor cells [46,47]) and (ii) the general cell types (endothelial cells [48] smooth muscle cells (SMCs) [49], and blood cells [50]). Moreover, SDF-1/CXCR4 signaling plays pivotal roles in cell chemotaxis, proliferation, migration, and differentiation that are considerably involved in embryonic development and stem/progenitor cell chemotaxis, as well as organ-specific homing in ischemic tissue and/or damaged tissue repairing [51,52]. In addition, upregulation of SDF-1/CXCR4 signaling is highly associated with tissue generation including myelopoiesis, cardiogenesis, angiogenesis, neurogenesis [53,54,55], and stem cell differentiation [56,57]. For instance, dysregulation of the SDF-1/CXCR4 axis would cause abnormal development of neural stem/progenitor cells in the cerebellum, hippocampus, and cortex [58]. In this study, we demonstrated that Au-decorated GO nanocomposites significantly elicited the activation of SDF-1/CXCR4 signaling and enhanced MMP2/9 expression in MSCs. Moreover, Au-decorated GO nanocomposites also effectively guided MSC-derived tissue differentiation, including neuron cells, endothelial cells, adipocytes, and osteocytes cells. Therefore, these results suggest that Au-decorated GO nanocomposites could serve as a potential bionanomaterial for ischemic tissue repair and tissue regeneration. 

Mesenchymal stem cells are multilineage differentiated stem cells, which can successfully induce connective tissue phenotypes (i.e., osteoblasts, adipocytes, and chondrocytes), neural tissues, and endothelial differentiation [8]. Here, we verified the effects of these as-prepared nanocomposite on mediated MSC differentiation. MSCs were cultured with nanomaterials for 7 days, and then subjected to specific markers that were associated with neurogenic differentiation by using an immunofluorescent assay (Figure 6). As compared with the control group, our results showed that neuron markers such as nestin, GFAP, and β-tubulin were significantly increased 7.0-, 13-, and 5.7-fold, in GO-Au (×2); increased 4.3-, 7.1-, and 5-fold in GO-Au; and increased 2.3-, 2.3-, and 3.3-fold in GO, respectively (Figure 6A–F). Furthermore, to demonstrate the as-prepared nanocomposites mediated endothelial differentiation, we analyzed the endothelial markers including CD31 and vWF expression profiles using an immunofluorescent assay (Figure 7). Our data revealed that GO, GO-Au, and GO-Au (×2) remarkably elicited upregulation with a 0.96-, 3.4- and 5.7-fold increase in CD31 expression (Figure 7A,B) and 3.4-, 7.6- and 13.1-fold increase in vWF expression, respectively (Figure 7C,D), as compared with the control group. These results indicate that Au-deposited GO nanocomposites, especially GO-Au (×2), are effective inducible nanocomposites for MSC differentiation as compared with the GO and the control groups. However, we also found that the nanomaterials mediated adipogenic and osteogenic differentiation abilities by using ORO staining and ARS staining, respectively, which were lower than neural and endothelial cell differentiation (Appendix A). 

### 3.5. In Vivo Biocompatibility and Anti-Inflammatory Effect of GO-Au

The greatest challenge for nanomaterial applications for tissue regeneration is an undesirable immune response to nanomaterial implanting, which would interfere with tissue repairing, induce incompatibility between nanomaterial devices and tissues, and eliciting a severe fibrotic formation. Finally, all the defects would lead to a failure of tissue regeneration [59]. Here, to validate the in vivo biocompatibility of Au-decorated GO nanocomposite implantation, an animal model was subcutaneously inoculated with as-fabricated nanomaterials for 4 weeks, and the effects of nanomaterial-mediated immune compatibility were further evaluated. As shown in Figure 8, first, we analyzed the effects of as-prepared nanomaterials on tissue fibrotic encapsulation. As compared with the control and GO groups, we found that Au-decorated GO nanocomposites significantly induced thinner collagen deposition and less fibrous capsule formation by using the Masson’s trichrome-stained assay (Figure 8A) and the H&E staining analysis (Figure 8B), respectively. Further quantification analysis indicated that both GO-Au and GO-Au (×2) efficiently decreased collagen deposition by 0.65-fold and 0.5-fold, respectively (Figure 8C), and significantly attenuated capsule thickness by 0.65-fold and 0.52-fold, respectively (Figure 8D), as compared with the control group and the GO group (a 0.87-fold and 0.82-fold in the collagen deposition and the capsule thickness, respectively.). We further confirmed the capacity of Au-decorated GO nanocomposites for tissue regeneration such as facilitated tissue angiogenesis. By using the immunochemical analysis, we detected the expression level of the endothelial marker CD31 on the tissues surrounding the implanted nanocomposites. Our data revealed that Au-decorated GO nanocomposites significantly elicited the upregulation of CD31 (increase of 1.25- and 1.32-fold in GO-Au and GO-Au (2×), respectively) as compared with the control groups (Figure 8E,F). This result suggests that the Au-decorated GO nanocomposites significantly induce endothelialization in vivo and could be an efficacy biomaterial for facilitating angiogenesis during tissue regeneration. Moreover, owing to the pivotal roles of macrophage polarization on biomaterial implant-mediated immune response, next, we detected the as-prepared nanomaterials on chronic inflammatory effects by analyzing the expression profiles of proinflammatory M1 macrophages and anti-inflammatory M2 macrophages. As shown in Figure 9A,B, by examining biomarkers of CD86 and CD163 on M1 and M2 macrophage, respectively, using an immunofluorescent assay, we observed that tissues surrounding the implanted Au-decorated GO nanocomposites exhibited lower expression levels of M1 macrophage (a decrease of 0.52- and 0.42-fold in GO-Au and GO-Au (×2), respectively (Figure 9C)), and exhibited higher expression levels of M2 macrophage (an increase of 1.36- and 1.44-fold in GO-Au and GO-Au (×2), respectively), as compared with both the control and GO groups (0.82- and 1.1-fold in M1 and M2 macrophages, respectively) (Figure 9D). These results suggest that the Au-decorated GO nanocomposites, especially GO-Au (×2), have excellent biocompatibility, inhibition of fibrotic formation, and lower immune stimulation in vivo, which all imply Au-decorated GO nanocomposites could be potential and ideal biomaterial for tissue regeneration. 

Foreign body reactions resulting from immune recognition mediate a cascade of cellular processes such as chronic inflammation, formation of foreign body giant cells, fibrosis, and damage to surrounding tissues, which, finally, cause tissue implantation failure [60]. In this study, the as-prepared Au-decorated GO nanocomposites showed properties of an ideal biomaterial for immunomodulatory effects by decreased immune responses including inhibition of macrophages activity, decreased platelet adhesion (Figure 3), and attenuation of ROS generation on MSCs (Appendix A). Moreover, animals implanted with Au-decorated GO nanocomposites also revealed lower foreign body reactions such as lower collagen deposition and less fibrous capsule formation. Indeed, the upregulation of M2 macrophages significantly induced by Au-decorated GO nanocomposites indicate the biocompatibility and anti-inflammatory effect of these nanocarriers and also suggest their enhancement of angiogenesis and tissue regeneration [61]. Additionally, an excellent bionanomaterial for tissue engineering should have good biocompatibility and also elicit an even distribution of nutrients in the implanted tissue. Vasculogenesis and angiogenesis are both critical processes for tissue regeneration and/or tissue engineering. The effective construction of vascular networks including venules, capillaries, and arterioles, can help nutrient and oxygen transport and support the stability and compatibly for repair or replacement of damaged tissues or organs [62]. Here, we demonstrated that Au-decorated GO nanocomposites significantly elicited MSC-endothelial cell differentiation (Figure 7) and successfully promoted endothelialization of surrounding tissue in the implanted animals (Figure 8E,F). 

Altogether, by fabricating biofunctional Au-decorated GO nanocomposites, we demonstrated they have good biocompatibility, induce MSCs-derived tissue differentiation including neuron cells, endothelial cells, adipocytes, and osteocytes, and have a lower immune response. The expression of CD68 was the lowest in GO-Au, following by GO and the control group (TCPS) in vitro (Figure 3). Subsequently, the markers of macrophage, CD86, and CD163, were selected as M1 and M2 polarization markers; the expression of CD86 was the lowest in GO-Au, following by GO and the control group, and the expression of CD163 was more prominent in GO-Au, following by GO and the control group (glass) (Figure 9). On the basis of these results, it is suggested that Au-deposited GO nanocomposites may be a potent nanocarrier for anti-inflammatory response due to foreign body reaction for biomaterial application. Moreover, we also validated that Au-decorated GO nanocomposites can decrease the host response during their implantation (Figure 10). Therefore, the as-fabricated Au-decorated GO nanocomposites can be considered as potent tissue engineering bionanomaterials.

The literature indicated that the scavenging of free radicals and antioxidant activity of metals such as gold nanoparticles can be utilized as free radical scavengers with biocompatibility [63]. Controlling the proinflammation induced by proinflammatory cytokines and the anti-inflammatory response induced by M2 macrophages is important for tissue repair. The M2 macrophage polarization presents a positive role for regeneration. AuNPs reduces cell damage by interfering with the NF-kB pathway and decreasing the levels of cytokines, such as IL-1β and TNF-α, as well as pro-apoptotic proteins [64]. AuNPs also reduce the NO and nitric oxide induced synthase (iNOS) levels, probably through regulating the transcription of the iNOS gene [65]. In addition, anti-inflammatory properties of AuNPs may be associated with reduced free radical levels as well as increased activity of antioxidant enzymes, since activated macrophages and neutrophils produce ROS [66,67]. Moreover, monocyte–platelet interactions may play a key role in this process by various pathways. These processes promote monocyte recruitment (in)to the vascular wall as a key mechanism in atherogenesis [68]. Platelets can recruit and stimulate monocytes by cytokines or by direct cell–cell interaction [69,70]. These characteristics may explain why graphene incorporated with AuNPs can lead to excellent cell viability and anti-oxidative ability, attenuate the immune response toward monocyte-macrophage transition, as well as inhibit the activity of the platelets in the current study.

## 4. Conclusions

In this study, we fabricated Au-deposited GO nanocomposites including GO-Au and GO-Au (×2), and methodologically analyzed their biological effects on the biocompatibility and anti-inflammatory response in vivo and in vitro. Our results showed that Au-deposited GO nanocomposites, especially GO-Au (×2), significantly exhibited excellent cell viability for MSCs, had good anti-oxidative ability, suppressed the immune response toward monocyte-macrophage transition, as well as inhibited the activity of the platelets. Moreover, we also validated the superior efficacy of Au-deposited GO nanocomposites on the enhancement of cell motility and MSC-derived various cell types of differentiation including neuron cells, adipocytes, osteocytes, and endothelial cells. Additionally, the lower induction of fibrotic formation, reduced M1 macrophage polarization, and higher induction of M2 macrophage, as well as promotion of the endothelialization, were also found in the Au-deposited GO nanocomposite-implanted animal model. These results suggested that the Au-deposited GO nanocarrier has excellent immune compatibility and anti-inflammatory effects in vivo and in vitro. Therefore, our findings indicated that the Au-decorated GO can be potential nanomaterials for biomedical applications. 

## Figures and Tables

**Figure 1 nanomaterials-11-02046-f001:**
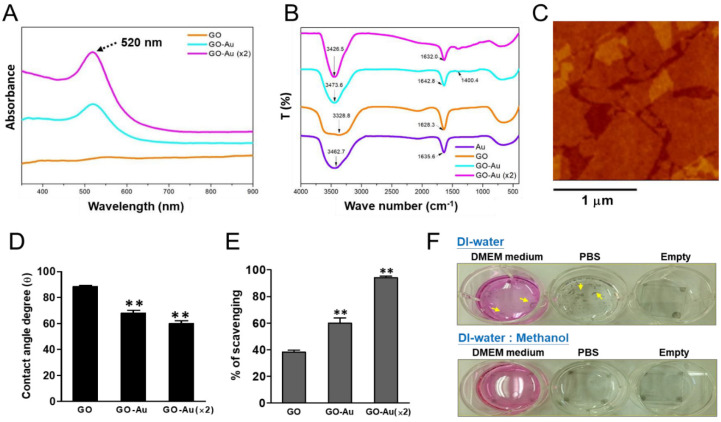
Characterization of as-fabricated Au-decorated GO nanocomposites. Three as-fabricated nanocomposites including GO, GO-Au, and GO-Au (×2) were subjected to analyze: (**A**) UV-Vis absorption spectra, where the peak at 520 nm is the absorption wavelength of AuNPs; (**B**) FTIR spectra; (**C**) AFM image of GO; (**D**) average contact angle (θ). The contact angle from different materials without water is θ = 0°. Finally, (**E**) trapping potential for DPPH radical scavenging activity of three as-fabricated nanocomposites was also validated. The data were expressed as means ± SD of three independent experiments. ** *p* < 0.01. To validate the GO effectively stabilized on the glass substrates, first, GO was well-dispersed in either (**F**) DI-water or DI-water/methanol (MeOH) mixture solution (1:1, *v*/*v*) and then spin-coated onto glass substrates (cover slides). Next, these samples were placed on culture plates and soaked in different solutions including DMEM culture medium, PBS, and without solution (empty). Small pieces of GO film were suspended (indicated as yellow arrows), whereas well-coated and stabilized GO film was found.

**Figure 2 nanomaterials-11-02046-f002:**
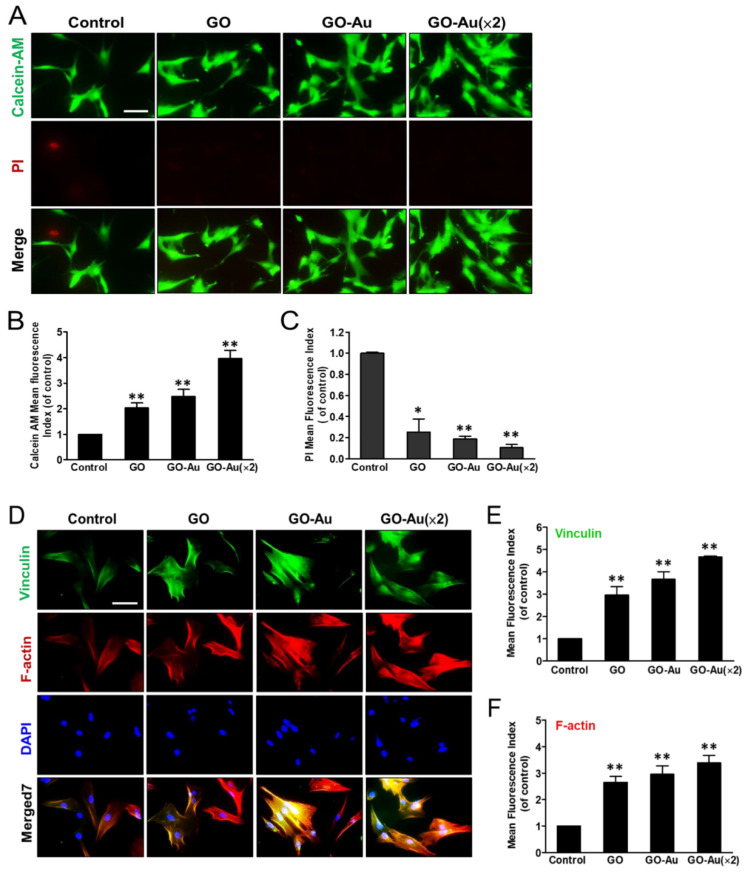
Effects of different materials on MSCs viability using calcein-AM/PI co-staining assay. Cells were cultured in cover slide deposited with nanomaterials including GO, GO-Au, and GO-Au (×2) for 24 h. Cells were, then, co-stained with calcein-AM dye and PI, and cell viability was observed using a fluorescence microscopy. (**A**) Both the live cells and dead cells were presented in green and red fluorescence images, respectively. The fluorescence intensity in either (**B**) calcein-AM (live cells) or (**C**) PI (dead cells) with further quantification and analysis using Image J software. The control group is indicated as a glass cover slide. Scale bars, 20 µm. Data were presented as the mean ± SD, * *p* < 0.05 and ** *p* < 0.001. Effects of different materials on MSCs adhesion ability and cell morphology: (**D**) After cells were cultured on various nanomaterials of GO, GO-Au, and GO-Au (×2) for 24 h, then, cells were analyzed for cell adhesion capacity using an immunofluorescence assay and recognized with anti-vinculin antibody (green) and cytoskeletal F-actin phalloidin dye (red). Nuclei were stained with DAPI (blue). The representative fluorescent intensity of (**E**) vinculin and (**F**) F-actin were determined from three independent results. Scale bar, 20 μm. Data were presented as the mean ± SD, ** *p* < 0.001.

**Figure 3 nanomaterials-11-02046-f003:**
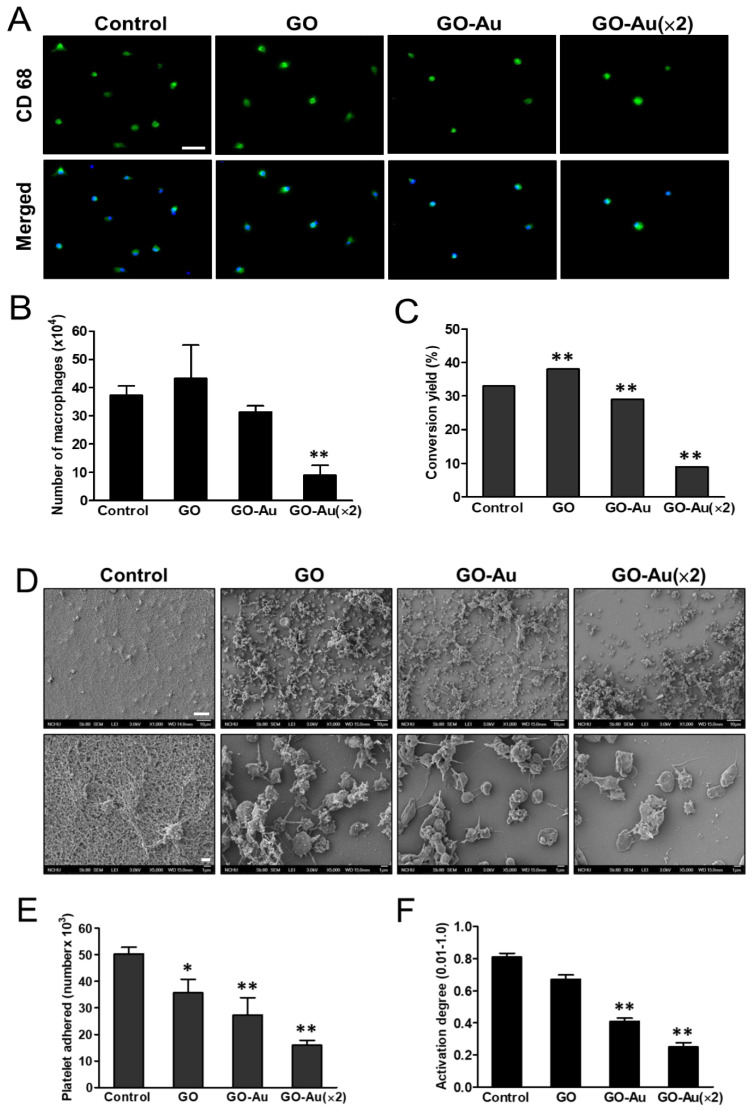
Effects of different materials on macrophage activation. Monocytes were isolated from whole blood and were incubated on various nanocomposites of GO, GO-Au, and GO-Au (×2) for 96 h. (**A**) Macrophage marker CD68 (green) was examined using an immunofluorescence staining. Cell nuclear was stained with DAPI (blue). The positive CD68-expressed cells were regarded as macrophages that were further counted and quantified for detection of macrophage activation (**B**) and monocyte/macrophage conversion rate (**C**). Scale bar = 20 μm. Data were presented as the mean ± SD, * *p* < 0.05 and ** *p* < 0.01. Effects of different materials on platelets activation. Platelets were obtained from human plasma and were incubated on various as-prepared nanocomposites at 37 °C for 60 min. (**D**) Platelet’s morphological alteration was further analyzed and recorded using scanning electron microscope (SEM) analysis. Scale bar = 10 μm (upper panels) and 1 μm (lower panels). Moreover, the as-prepared nanocomposite-mediated platelet adherence efficiency (**E**) and degree of platelet activation (**F**) were further calculated and quantified based on three independent SEM images. The data were expressed as means ± SD. * *p* < 0.05 and ** *p* < 0.01.

**Figure 4 nanomaterials-11-02046-f004:**
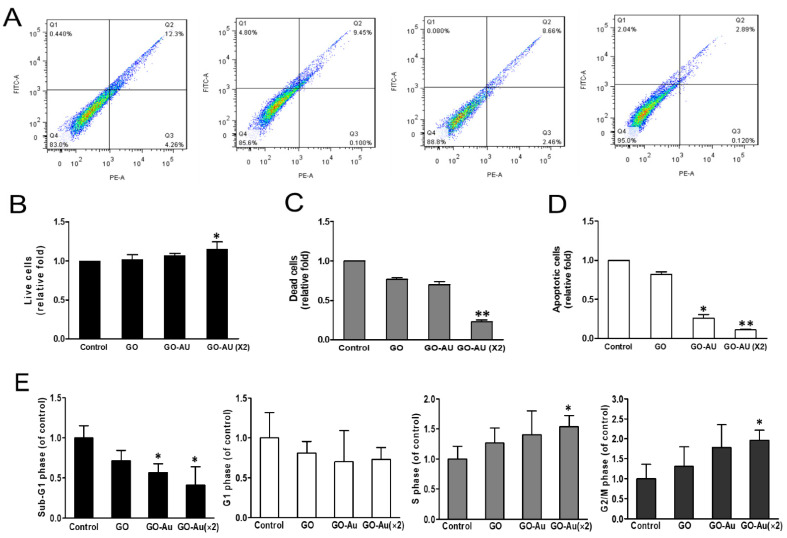
The cell viability assay of MSCs on different materials. (**A**) Annexin V-PI double staining was used by flow cytometry. The results evaluated that the number of (**B**) live cells, (**C**) dead cells, (**D**) apoptotic cells. * *p* < 0.05 and ** *p* < 0.01. Cells were cultured on various nanomaterials of GO, GO-Au, and GO-Au (×2) for 24 h, and then cells were harvested and subjected to analyze cell cycle distribution using a flow cytometry analysis. (**E**) The cell cycle distribution was further analyzed and shown in bar graphs. The quantification in respective cell cycle phases of Sub-G1, G1, S, and G2/M. All values were expressed as means ± SD of three independent experiments. * *p* < 0.05.

**Figure 5 nanomaterials-11-02046-f005:**
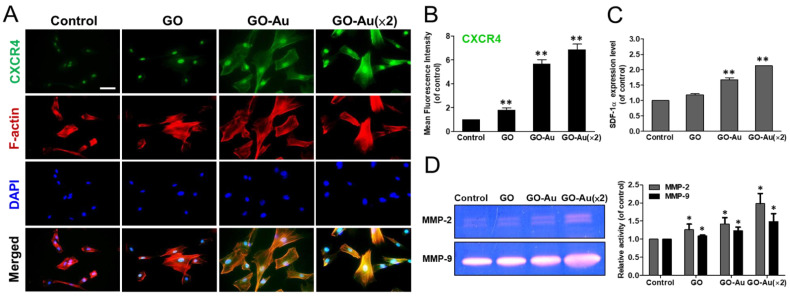
Biological assessment of MSCs on different materials. After cells were cultured on various nanomaterials of GO, GO-Au, and GO-Au (×2) for 24 h and/or 48 h, (**A**) cells were, then, analyzed for cell migratory ability using an immunofluorescence assay through incubation with anti-CXCR4 antibody (green) and F-actin phalloidin dye (red). Scale bar = 20 μm. (**B**) The quantification data obtained from the fluorescence intensity of CXCR4 expression. (**C**) The expression level of SDF-1α and (**D**) the MMPs’ activity were obtained from supernatant of MSCs grown on various nanomaterials for 48 h using ELISA analysis and gelatin zymography analysis, respectively. The quantification of MMP-2/9 expression was also acquired using an Image Pro Plus 5.0 software. Data were presented as the mean ± SD, * *p* < 0.01 and ** *p* < 0.001.

**Figure 6 nanomaterials-11-02046-f006:**
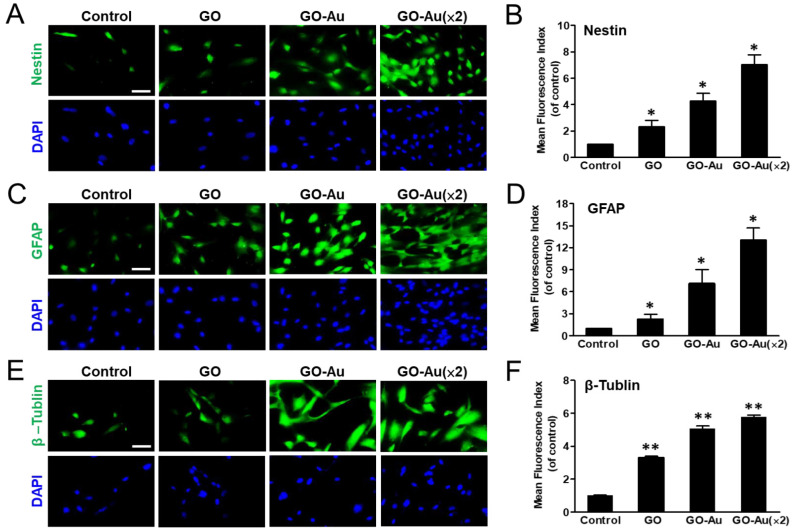
Characterization of GO-Au-mediated MSC differentiation into neuron cells. Cells were cultured on various nanomaterials of GO, GO-Au, and GO-Au (×2) for 7 days. Then, cells were analyzed for expression levels of (**A**,**B**) nestin, (**C**,**D**) GFAP and (**E**,**F**) β-tubulin, using an immunofluorescence assay. Scale bar, 20 μm. The quantification of fluorescent intensity in nestin, GFAP, and β-tubulin protein expression. Data were presented as the mean ± SD, * *p* < 0.05 and ** *p* < 0.01.

**Figure 7 nanomaterials-11-02046-f007:**
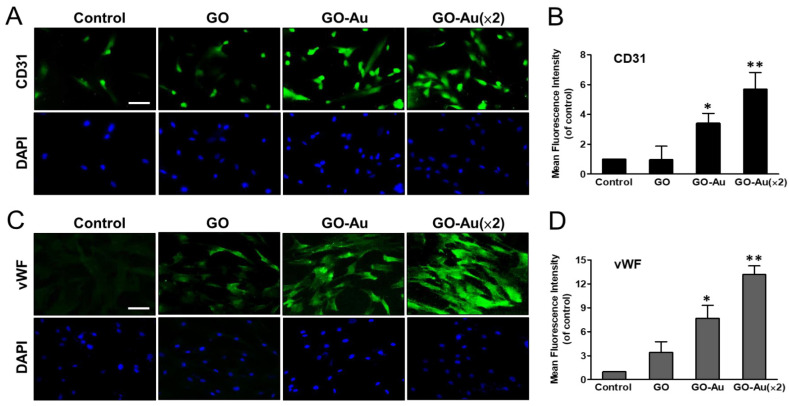
Identification of GO-Au-mediated MSC differentiation into endothelial cells. Cells were cultured on various nanomaterials of GO, GO-Au, and GO-Au (×2) for 7 days. Then, cells were analyzed for the expression levels of (**A**,**B**) CD31 and (**C**,**D**) vWF, using an immunofluorescence assay. Scale bar, 20 μm. The quantification of fluorescent intensity in CD31 and vWF protein expression. Data were presented as the mean ± SD. * *p* < 0.05 and ** *p* < 0.01.

**Figure 8 nanomaterials-11-02046-f008:**
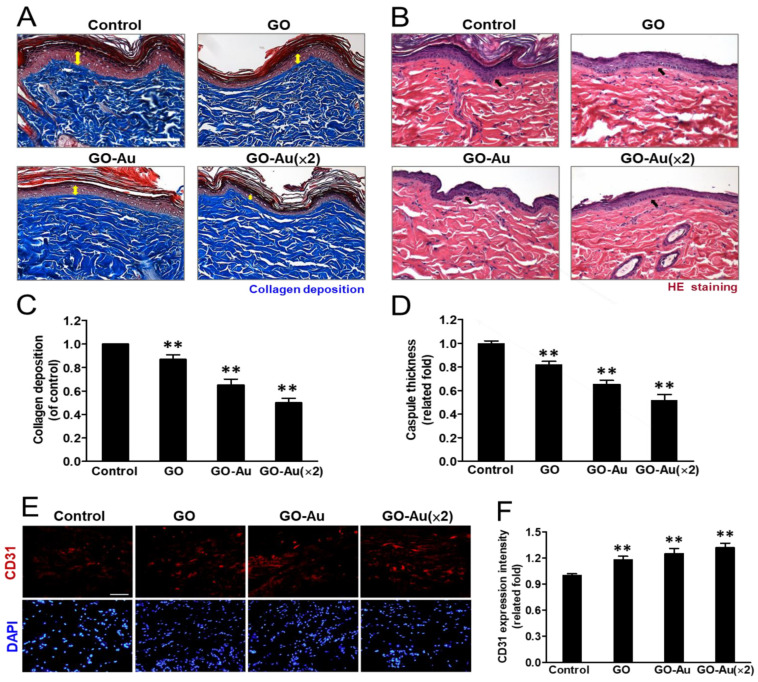
Histological analysis of foreign-body response (FBR) of different materials following four-week subcutaneous implantation. Photomicrographs showing (**A**) the Masson’s trichrome-stained histological sections and (**B**) the hematoxylin and eosin (H&E)-stained histological sections of as-prepared nanocomposites. (**C**) Quantification of collagen deposition (yellow double arrow in panel A) on the histology examination. (**D**) Analysis of the FBR was revealed with the capsule thickness (arrows in panel B) based on the histology examination. Scale bar = 100 μm. Data were presented as the mean ± SD (*n* = 6). ** *p* < 0.01 as compared with the control group. Immunohistochemical analysis of endothelialization. (**E**) Immunofluorescent staining of explanted nanocomposites of GO, GO-Au, and GO-Au (×2) with surrounding tissue were stained for endothelialization marker of CD31. Cell nuclei were stained with DAPI (blue). Scale bar = 20 μm. (**F**) The fluorescence intensity of CD31 was quantified and recorded. Data were presented as the mean ± SD (*n* = 6). ** *p* < 0.01 as compared with the control group.

**Figure 9 nanomaterials-11-02046-f009:**
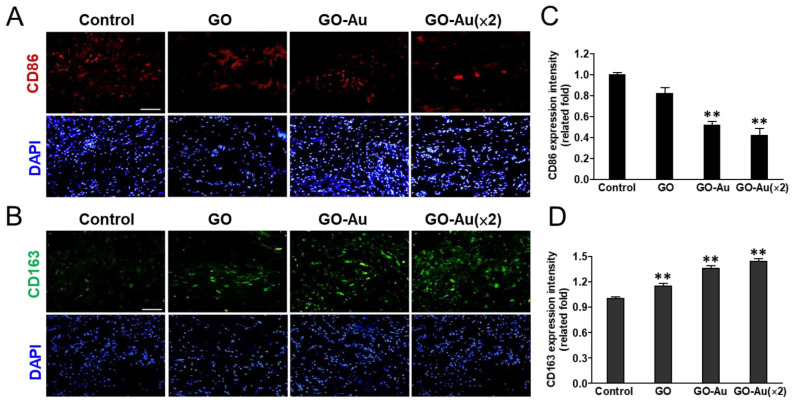
Immunohistochemical analysis of macrophage phenotype markers. Sections of explanted nanocomposites of GO, GO-Au, and GO-Au (×2) with surrounding tissue were stained for macrophage phenotype markers of (**A**) CD86 (M1) (red) and (**B**) CD163 (M2) (green). Cell nuclei were stained with DAPI (blue). Scale bar = 20 μm. The fluorescence intensity of (**C**) CD86 and (**D**) CD163 were quantified and recorded. Data are shown as mean ± SD (*n* = 6). ** *p* < 0.01 as compared with the control group.

**Figure 10 nanomaterials-11-02046-f010:**
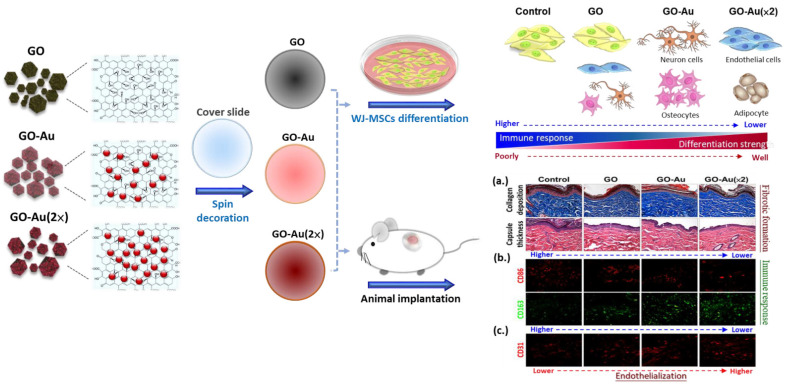
The schematic diagram illustrated that Au-deposited GO nanocomposites elicit MSC differentiation and attenuate immune response in vitro and in vivo.

## Data Availability

Data is contained within the article.

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
