# Peer review of "Nanogold-Carried Graphene Oxide: Anti-Inflammation and Increased Differentiation Capacity of Mesenchymal Stem Cells"

_nanomaterials, 2021, doi:10.3390/nano11082046_

Round 1
Reviewer 1 Report
1. What was the concentration of GO used for layer deposition? What was the GO concentration per cm2 of the surface (for ex. g/cm2) or what was the thickness of the formed GO layer (AFM measurements will be helpful).
2. The square glass slides were placed in round well-plates. This gives some risk in the evaluation of biological tests because the cells attach not only to square slides but also to a quiet large area of PS well-plates. It will be better to use round slides with a proper diameter.
3. In the Fig. 1 there are 3 spectra and one of them is assigned as „GO”. I think that it is rather spectrum of Au, not GO. The spectrum of pure GO should be also included.
4. SEM or TEM pictures of GO and GO with Au (also Au2x) are necessary. It is very important for this work to show the size, shape and distribution of Au particles deposited on GO flakes. Also the information about the size of GO flakes should be included.
5. The temperature of Au ions reduction was 90 C. At this temperature GO can be also reduced and some part of oxygen groups can be removed. Please show and discuss the difference in GO structure before and after this process. Some peaks on FTIR spectra may suggest the changes.
6. In experimanteal section the authors write the following sentence: „…nanocomposites were executed and characterized by following our previously study.26, 27”. Is the reference 27 correct? I do not see any connections to the authors of this cited work.
7. I would like to point out that the figures of the cells’ imaging are of a very good quality and the number of biological tests is large. This is the strong point of this maunscript.
Author Response
Reviewer 1:
- What was the concentration of GO used for layer deposition? What was the GO concentration per cm2 of the surface (for ex. g/cm2) or what was the thickness of the formed GO layer (AFM measurements will be helpful).
Answer:
Thanks the valuable comment from the reviewer. We have included the AFM image into the new Figure 1C and included the description in the “Materials and Methods” section “The surface morphology of the GO films was obtained using a Digital Instruments Dimension 3100 atomic force microscope.” (Page 5, line 23-24); in the “Results” section “For pure GO, the concentration was approximately 0.01 mg mL-1. The AFM image is shown below. We observed that the GO covered more than 80% of the surfaces. The average size of the GO flakes was 2 mm, and the calculated surface concentration was 2´103 cm-2. Without Au nanoparticles, the average thickness of the GO planes was 1.5 nm, which is the value for one or two layers of GOs (Figure 1C).” (Page 12, line 13-16)
- The square glass slides were placed in round well-plates. This gives some risk in the evaluation of biological tests because the cells attach not only to square slides but also to a quiet large area of PS well-plates. It will be better to use round slides with a proper diameter.
Answer:
We agreed with the consideration by reviewer there may be a small area of the edge that cannot be fully covered by the materials coated in the square slides. However, GO-Au demonstrated a superior biological efficacy than GO and the TCPS control group in the study, suggesting the current protocol is still appropriate.
- In the Fig. 1 there are 3 spectra and one of them is assigned as „GO”. I think that it is rather spectrum of Au, not GO. The spectrum of pure GO should be also included.
Answer :
We have performed and replaced the new FTIR data and included the GO group in Figure 1B.
- SEM or TEM pictures of GO and GO with Au (also Au 2x) are necessary. It is very important for this work to show the size, shape and distribution of Au particles deposited on GO flakes. Also the information about the size of GO flakes should be included.
Answer:
(1) We have included the more detail description in the “Results and Discussion” section “The shape of the AuNP was very close to spherical and the size of the AuNP was 7 nm in average. The distribution AuNPs was somehow not even. We found that more AuNPs formed on the edges and at the wrinkles, presumably because of the higher numbers of defect sites on these areas [30].” (Page 12, line 16-19).
(2) Because the materials we used were the same as those we published before [J. Nanomater. 2014, 736879 (2014)], which is also the reference [30]. The relevant information could be found in the paper. Further, the average size of the GO flakes was 2 mm, as we indicated in the respone to Question 1.
- The temperature of Au ions reduction was 90 C. At this temperature GO can be also reduced and some part of oxygen groups can be removed. Please show and discuss the difference in GO structure before and after this process. Some peaks on FTIR spectra may suggest the changes.
Answer:
We thanks the valuable comment from the reviewer. We have performed the new FTIR data and description in the “Results and Discussion” section “By using FTIR analyses as shown in Figure 1B, The spectra of GO revealed broad absorption bands near 3328.8 cm−1 and 1628.3 cm−1, which were attributed to the stretching mode of O–H bond [35] and C-O vibration, respectively [36]. The naked AuNP displayed signals at 3462.7 cm−1 and 1635.6 cm−1 [37]. The strong peak at around 3400 cm−1 is due to the stretching mode of –OH [38]. When AuNPs were incorporated onto GO, the shifted adsorption peaks at 1642.8 cm-1 (GO-Au) and 1632.0 cm-1 [(GO-Au (x2)] were observed. The new peak located at 1425 cm-1 of GO-Au and GO-Au (x2) was ascribed to the carboxyl O=C–O or –OH bond from the –COOH sensing group in the functionalized GO-Au [38, 39].” (Page 12, line 5-12)
- In experimanteal section the authors write the following sentence: „…nanocomposites were executed and characterized by following our previously study.26, 27”. Is the reference 27 correct? I do not see any connections to the authors of this cited work.
Answer:
We have changed the wording “by following our previously study” to “as in the previous report”. (Page 5, line 4)
- I would like to point out that the figures of the cells’ imaging are of a very good quality and the number of biological tests is large. This is the strong point of this maunscript.
Answer:
We appreciate your comment.

Reviewer 2 Report
The article can be accepted for publication after some changes.
In the introduction should be added the additional paragraph explained the behavior of cell culture on the graphene surfaces.
Would you please explain the abbreviation GO-Au (×2)? Authors used (×2) and (2x). One style should be selected.
Why do the authors use lyophilization? "Finally, both the GO-Au and GO-Au (2x) nanocomposites were dried through lyophilization."
In Fig 1. the authors demonstrate the absorption spectra for GO and composite material. Why is the peak 520 present in GO materials? I think it is a mistake with an abbreviation. It is better to normalize the absorption spectra.
Would you provide the information about the measurement FTIR spectra? As I can understand, it is recorded in the transmission mode.
Could you explain the value of contact angle is 88? It is the typical value for graphite and graphene. Did the authors use the rGO? The GO has a hydrophilic nature. It is nice to add the Raman spectra of the films or XPS analysis.
Change the scale bar in Fig 3 D.
Please try to speculate why adding the gold leads to excellent cell viability, well anti-oxidative ability, and sponged the immune response toward monocyte-macrophage transition as well as inhibited the activity of the platelets?
Author Response
Reviewer 2:
The article can be accepted for publication after some changes.
- In the introduction should be added the additional paragraph explained the behavior of cell culture on the graphene surfaces.
Answer:
We have included the more detail description in the “Introduction” section.
(1) “An adequate substrate and/or scaffold for tissue engineering is to provide several critical properties such as mechanical support, chemical stimuli, and biological signals for induction of cell adhesion and proliferation [24, 25]. The advantage of GO-Au may be ascribed to the abundant surface chemical groups on the GO and/or AuNPs deposition as well as the the enhanced surface topography, which are associated with cell adhesion and cell proliferation further. Indeed, surface features of nanocomposites including roughness, curvature, and wrinkled morphology all affect the cell adhesion and proliferation. For example, the 3D graphene foam with a wrinkled surface has been shown to induce neural cell growth rather than a smoother 2D graphene surface [26]. Moreover, the nanotopographic network features on the surface of graphene nanogrid patterns were demonstrated to successfully enhance the osteogenic differentiation of MSCs [25].” (Page 3, line 21-28; Page 4, line 1).
(2) “The potent application of GO-decorated with biopolymer for tissue engineering has been demonstrated with accumulating documents [29]. However, the efficacy of GO functionalized with metal nanoparticles such as silver and gold on stem cell-derived tissue engineering is still unclear and needs to be validated. In this study, we evaluated the effects of the as-fabricated AuNPs-decorated GO nanocomposites for the biocompatibility and cell adherent efficiency of MSCs.” (Page 4, line 4-8)
- Would you please explain the abbreviation GO-Au (2×)? Authors used (×2) and (2x). One style should be selected.
Answer:
We have corrected the abbreviation of GO-Au (2×) and make it to be consistent.
- Why do the authors use lyophilization? "Finally, both the GO-Au and GO-Au (2x) nanocomposites were dried through lyophilization."
Answer:
We have modified and wording the preparation procedure of GO-Au nanocomposites and made it to be clear and easier to follow in the “Materials and Methods” section “Graphene oxides (GO) and HAuCl4 and sodium citrate were purchased from Sigma-Aldrich. The synthesized procedure of GO-Au followed the Hummers’ method as in the previous report [30, 31]. GO-Au solution was prepared by mixing GO (0.275 mg/ml, 3 ml) and HAuCl4 (0.047 mg/ml, 10 ml) components and then aged for 30 min. After heating the solution to 80oC, sodium citrate (0.085 mol/dm3) was added to reduce gold ions for 4 hours of reaction. The as-prepared suspensions were then centrifuged (6000 rpms) and washed with ddH2O water twice to remove the unreacted gold ions. Ultimately, the lyophilization was processed to acquire dry GO-Au nanocomposites.” (Page 5, line 3-9).
- In Fig 1. the authors demonstrate the absorption spectra for GO and composite material. Why is the peak 520 present in GO materials? I think it is a mistake with an abbreviation. It is better to normalize the absorption spectra.
Answer:
Thanks for the valuable comment from the reviewer. We have provided the new data and corrected Figure 1A.
- Would you provide the information about the measurement FTIR spectra? As I can understand, it is recorded in the transmission mode.
Answer:
Thanks for the valuable comment from the reviewer. We have included the more detail description in the “Materials and Methods” section “Fourier transform infrared (FTIR) spectra were acquired (8 cm−1resolution 256 scans with the sample compartment vacuum pressure set at 0.12 hPa) using a Bruker 66 v/s FTIR spectrometer. Double sided polished silicon (100) wafer substrates were cut into 20 × 20 mm2 pieces with a diamond-tipped stylus. Spectra from a freshly plasma-cleaned silicon wafer sample were collected before each measurement to obtain the background spectrum.” (Page 5, line 18-22)
- Could you explain the value of contact angle is 88? It is the typical value for graphite and graphene. Did the authors use the rGO? The GO has a hydrophilic nature. It is nice to add the Raman spectra of the films or XPS analysis.
Answer:
We indeed used GOs as starting materials. The same materials were used for other photovoltaic applications. More characteristics can be found in our previous publication [J. Nanomater. 2014, 736879 (2014)] [30]. From the Raman spectra published previously [30] (Figure 3 in the paper, and is also shown below), the ratio of the intensities of the D and G bands (?D/?G) is often used to measure the extent of disorder. The ratio of GO was 0.91; and it decreased slightly to 0.90 for AuNP-GO/GL composites. Therefore, we suspected that the GO nanosheets were reduced only slightly during the reduction processes of the Au ions. Additionally, we treated the substrates with UV-ozone before spin-coating the AuGO solution. We suspected that the rather hydrophilic faces of the AuGO may prefer the substrate side, exposing the rather hydrophobic face to the top. This was probably the reason for the contact angle of 88.
Figure 3. Raman spectra of GO and the nanocomposites.
- Change the scale bar in Fig 3 D.
Answer:
We have included the scale bar in Figure 3A and Figure 3D.
- Please try to speculate why adding the gold leads to excellent cell viability, well anti-oxidative ability, and sponged the immune response toward monocyte-macrophage transition as well as inhibited the activity of the platelets?
Answer:
We thank the valuable comment from the reviewer. We have included the more detail description in the “Results and Discussion” section “Literature indicated that the scavenging of free radicals and antioxidant activity of metals such as gold nanoparticles make them to be utilized as free radical scavengers with biocompatibility [63]. Controlling the pro-inflammation induced by pro-inflammatory cytokines and the anti-inflammatory response induced by M2 macrophages is important for tissue repair. The M2 macrophage polarization presents a positive role for regeneration. AuNPs, which reduces cell damage by interfering with the NF-kB pathway and decreasing the levels of cytokines, such as IL-1β and TNF-α, as well as pro-apoptotic proteins [64]. AuNPs also reduce the NO and nitrc oxide induced sintase (iNOS) levels probably through regulating the transcription of the iNOS gene [65]. In addition, anti-inflammatory properties of AuNPs may be associated with reduced free radical levels as well as the increased activity of antioxidant enzymes since activated macrophages and neutrophils produce ROS [66, 67]. Moreover, monocyte-platelet interactions may play a key role in this process by various pathways. These processes promote monocyte recruitment (in)to the vascular wall as a key mechanism in atherogenesis [68]. Platelets can recruit and stimulate monocytes by cytokines or by direct cell-cell interaction [69, 70]. These characteristics may explain why graphene incorporate with AuNPs can lead to excellent cell viability and anti-oxidative ability, attenuate the immune response toward monocyte-macrophage transition, as well as inhibit the activity of the platelets in the current study.” (Page 19, line 24-28) (Page 20, line 1-10)
Reviewer 3 Report
General comments:
The authors present evidence that GO-Au has benefitial properties like anti-oxidative ability or enhancement of MSC differentiation indication favorable immune capacity and anti-inflammatory effects. The study design combining in vivo and in vitro experiments is conceptualised well and described in great detail. I think the authors also did a great job in clearly presenting the results and carefully interpreting them. However, I find that the text is sometimes hard to read due to several language/styling mistakes (e.g. paragraph 3.2., p. 13 double "therefore") and would profit from further proofreading. Also, the introduction is extremly superficial and should be more specific and focus on Au NP and GO. Many specific information are given in the reults section, but I think some of them shold be removed to the introduction to give the reader a better inside into the current state of research. I really like your study but think some improvements in style might be needed
Minor comments:
Figure 1: I´d suggest you also comment on the wavelenght of 3433.75 in the text, as all other wavelenghts are mentioned.
Sometimes you use Au NP and sometimes AuNP.
Author Response
Reviewer 3:
General comments:
The authors present evidence that GO-Au has benefitial properties like anti-oxidative ability or enhancement of MSC differentiation indication favorable immune capacity and anti-inflammatory effects. The study design combining in vivo and in vitro experiments is conceptualised well and described in great detail. I think the authors also did a great job in clearly presenting the results and carefully interpreting them.
- However, I find that the text is sometimes hard to read due to several language/styling mistakes (e.g. paragraph 3.2., p. 13 double "therefore") and would profit from further proofreading.
Answer:
We thank for the valuable comment from the reviewer. We have corrected the typo (Page 14, line 6) and carefully proofread this manuscript again.
- Also, the introduction is extremly superficial and should be more specific and focus on AuNP and GO.
Answer:
We have included more detail description on AuNP and GO in the “Introduction” section.
(1) “An adequate substrate and/or scaffold for tissue engineering is to provide several critical properties such as mechanical support, chemical stimuli, and biological signals for induction of cell adhesion and proliferation [24, 25]. The advantage of GO-Au may be ascribed to the abundant surface chemical groups on the GO and/or AuNPs deposition as well as the the enhanced surface topography, which are associated with cell adhesion and cell proliferation further. Indeed, surface features of nanocomposites including roughness, curvature, and wrinkled morphology all affect the cell adhesion and proliferation. For example, the 3D graphene foam with a wrinkled surface has been shown to induce neural cell growth rather than a smoother 2D graphene surface [26]. Moreover, the nanotopographic network features on the surface of graphene nanogrid patterns were demonstrated to successfully enhance the osteogenic differentiation of MSCs [25].” (Page 3, line 21-28; Page 4, line 1).
(2) “The potent application of GO-decorated with biopolymer for tissue engineering has been demonstrated with accumulating documents [29]. However, the efficacy of GO functionalized with metal nanoparticles such as silver and gold on stem cell-derived tissue engineering is still unclear and needs to be validated. In this study, we evaluated the effects of the as-fabricated AuNPs-decorated GO nanocomposites for the biocompatibility and cell adherent efficiency of MSCs.” (Page 4, line 4-8)
- Many specific information are given in the reults section, but I think some of them shold be removed to the introduction to give the reader a better inside into the current state of research. I really like your study but think some improvements in style might be needed.
Answer:
We thank for the valuable comment from the reviewer. We have removed some result section into the “Introduction” section.
(1) “An adequate substrate and/or scaffold for tissue engineering is to provide several critical properties such as mechanical support, chemical stimuli, and biological signals for induction of cell adhesion and proliferation [24, 25]. The advantage of GO-Au may be ascribed to the abundant surface chemical groups on the GO and/or AuNPs deposition as well as the the enhanced surface topography, which are associated with cell adhesion and cell proliferation further. Indeed, surface features of nanocomposites including roughness, curvature, and wrinkled morphology all affect the cell adhesion and proliferation. For example, the 3D graphene foam with a wrinkled surface has been shown to induce neural cell growth rather than a smoother 2D graphene surface [26]. Moreover, the nanotopographic network features on the surface of graphene nanogrid patterns were demonstrated to successfully enhance the osteogenic differentiation of MSCs [25].” (Page 3, line 21-28; Page 4, line 1).
(2) “The potent application of GO-decorated with biopolymer for tissue engineering has been demonstrated with accumulating documents [29]. However, the efficacy of GO functionalized with metal nanoparticles such as silver and gold on stem cell-derived tissue engineering is still unclear and needs to be validated. In this study, we evaluated the effects of the as-fabricated AuNPs-decorated GO nanocomposites for the biocompatibility and cell adherent efficiency of MSCs.” (Page 4, line 4-8)
Minor comments:
- Figure 1: I´d suggest you also comment on the wavelenght of 3433.75 in the text, as all other wavelenghts are mentioned.
Answer:
We thanks the valuable comment from the reviewer. We have performed the new FTIR data and description in the “Results and Discussion” section “By using FTIR analyses as shown in Figure 1B, The spectra of GO revealed broad absorption bands near 3328.8 cm−1 and 1628.3 cm−1, which were attributed to the stretching mode of O–H bond [35] and C-O vibration, respectively [36]. The naked AuNP displayed signals at 3462.7 cm−1 and 1635.6 cm−1 [37]. The strong peak at around 3400 cm−1 is due to the stretching mode of –OH [38]. When AuNPs were incorporated onto GO, the shifted adsorption peaks at 1642.8 cm-1 (GO-Au) and 1632.0 cm-1 [(GO-Au (x2)] were observed. The new peak located at 1425 cm-1 of GO-Au and GO-Au (x2) was ascribed to the carboxyl O=C–O or –OH bond from the –COOH sensing group in the functionalized GO-Au [38, 39].” (Page 12, line 5-12)
- Sometimes you use Au NP and sometimes AuNP.
Answer:
We have corrected the Au NP to “AuNP” and made the abbreviation more consistent.

Reviewer 4 Report
This is a well constructed paper which includes the analysis of numerous parameters which support the aim of the study, but it needs an extensive English revision since a lot of results could be not well understood from a reader not expert in this field
Author Response
Reviewer 4:
This is a well constructed paper which includes the analysis of numerous parameters which support the aim of the study, but it needs an extensive English revision since a lot of results could be not well understood from a reader not expert in this field
Answer: We have proofread the article to make it well understood by the readers.
